# The Use of UVC Irradiation to Sterilize Filtering Facepiece Masks Limiting Airborne Cross-Infection

**DOI:** 10.3390/ijerph17207396

**Published:** 2020-10-11

**Authors:** Wojciech Kierat, Weronika Augustyn, Piotr Koper, Miroslawa Pawlyta, Arkadiusz Chrusciel, Bernard Wyrwol

**Affiliations:** 1Department of Digital Systems, Silesian University of Technology, 44-100 Gliwice, Poland; bernard.wyrwol@polsl.pl; 2Department of Environmental Biotechnology, Silesian University of Technology, 44-100 Gliwice, Poland; weronika.augustyn@mexeo.pl; 3Department of Heating, Ventilation and Dust Removal Technology, Silesian University of Technology, 44-100 Gliwice, Poland; piotr.koper@polsl.pl; 4Department of Engineering Materials and Biomaterials, Silesian University of Technology, 44-100 Gliwice, Poland; miroslawa.pawlyta@polsl.pl; 5MEXEO Institute of Technology, 47-225 Kedzierzyn-Kozle, Poland; arkadiusz.chrusciel@mexeo.pl

**Keywords:** ultraviolet germicidal irradiation, UVGI, HEPA filter sterilization, SARS-CoV-2

## Abstract

In addition to looking for effective drugs and a vaccine, which are necessary to save and protect human health, it is also important to limit, or at least to slow, the spread of coronavirus. One important element in this action is the use of individual protective devices such as filtering facepiece masks. Currently, masks that use a mechanical filter, such as a HEPA (High Efficiency Particulate Air) filter, are often used. In some countries that do not have a well-developed healthcare system or in exceptional situations, there is a real and pressing need to restore filters for reuse. This article presents technical details for a very simple device for sterilization, including of HEPA polymer filters. The results of biological and microscopic tests confirming the effectiveness of the sterilization performed in the device are presented. The compact and portable design of the device also allows its use to disinfect other small surfaces, for example a small fragment of a floor, table, or bed.

## 1. Introduction

Ongoing in every corner of the world, the fight against the CoVID19 virus pandemic presents scientists with the challenge of seeking an effective drug or vaccine to either eliminate, or at least significantly reduce, the growing disease caused by the pathogen. Within the wide range of actions taken in view of the threat of a pandemic, solutions aimed at limiting or slowing down the speed of spreading of the virus play an important role. In the opinion of virology specialists, it is advisable to use personal protective equipment such as filtering face masks. In some countries, the use of such masks is not only a recommendation but even a legally sanctioned order.

Unfortunately, the availability of protective filtering facepiece masks with sufficiently good filtration properties is limited. Unavailability of masks has been signaled by medical staff who are providing help to infected people. Considering the lack of a viable possibility of very quickly and sufficiently fully equipping medical services with personal protective devices, including filtering facepiece masks equipped with effective air filters, a good solution is to restore filters for use by sterilization. Extended use or re-use of filtering facepiece masks is not recommended by the World Health Organization (WHO), European Centre for Disease Prevention and Control (ECDC), the US Centers for Disease Control and Prevention (CDC), and Public Health England (PHE). However, in exceptional circumstances, it is allowed to depart from this rule under certain conditions [1]. The most important requirements are that the mask is not mechanically damaged, is not dirty (blood, secretions, body fluids), and that the sterilization process is carried out by properly trained personnel. The mentioned solution should be treated as a last-resort measure.

One of the possible methods of sterilizing air filters is exposure to ultraviolet radiation (UVGI—ultraviolet germicidal disinfection) with a wavelength range from 100 to 280 nm (UVC). The effectiveness of this sterilization method is confirmed by studies [2] in which severe acute respiratory coronavirus syndrome (SARS-CoV) and Middle East respiratory syndrome coronavirus (MERS-CoV) were inactivated. The method is based on inactivation of the coronavirus by damaging deoxyribonucleic acid, which prevents the reproduction of pathogen cells. Looking at the general similarities between the currently spreading SARS-CoV-2 and SARS-CoV and MERS-CoV, it can be presumed that UVGI is also effective in deactivating SARS-CoV-2. The problem of deactivating the pathogen by UVC radiation can be also found in many recent publications [3,4,5,6]. The primary purpose of our research is to develop the device and determine its effectiveness. The possibility of using the UVGI method has recently been proposed and generally described [7]. However, no specific calculations or sufficient technical details have been presented, which are needed in order to construct such a device.

The main issue in the design of the device was to determine the required minimum exposure time to disinfect the surface, so as to effectively deactivate the pathogen-containing particles, as well as the maximum exposure time, exceeding which could cause permanent damage to the filter surface. As demonstrated in previous studies [6,8,9], a sterilization dose of approximately 1 J/cm² is sufficient to sterilize the surface of the material. It has also been shown [8,10] that increasing the radiation dose above 1 J/cm² does not bring significant benefits. Polymers are often used for the production of air filters and a high dose of ultraviolet radiation may result in a deterioration of their filtering properties [11]. The degree of sensitivity of the filter to UV radiation depends on the quality and technology of the particular polymer. Test results showed that for some polymers, their properties deteriorated by 1% when exposed to a radiation dose of 100 J/cm² [11,12]. Determining the exposure time based on the assumed range of radiation dose variability requires taking into account the power and spatial distribution of the ultraviolet radiation and the distance of the sterilized object from the radiator or radiators.

Sterilization is a process aimed at the complete destruction and elimination of all living microorganisms and their spore forms. The use of biological indicators is considered the most reliable, easy, and quick method of sterilization process control. The indicators are described in the EN ISO 11138-1: 2017 standard, concerning the sterilization of products in health care [13]. The biological indicator system contains a population of spores of a non-pathogenic microorganism, appropriate for a given type of sterilization, with the highest resistance to the action of a particular sterilizing agent. Microorganisms are most often placed on paper strips and stainless-steel discs [14,15].

Microorganisms vary widely in their resistance to sterilization and/or disinfection. With the exception of prions, bacterial spores have the greatest innate immunity. Subsequently, the relative scale of resistance includes mycobacteria, non-enveloped viruses, Gram-positive bacteria, fungi, and Gram-negative bacteria. The envelope viruses, which include the coronaviruses, are the most sensitive to disinfection processes, see Figure 1 [16,17].

In the case of using UVC radiation, it has been shown that the elimination of bacterial spores requires a radiation dose 3 times higher than for the inactivation of polio- and rotaviruses [18]. In turn, enveloped viruses, under the influence of reactive oxygen species induced by UVC radiation, are more susceptible to inactivation than viruses without envelopes. The lipids building the envelope undergo peroxidation [19]. Therefore, it can be concluded that the dose of radiation eliminating bacterial spores will also be destructive to enveloped viruses.

Due to their small size and ease of use, paper indicator strips containing *Bacillus atrophaeus* ATCC 9372 (the bacterial strains of *B. atrophaeus* from American Type Culture Collection, global biological materials resource and standards organization, no 9372) spores were selected for testing. Although the strain is dedicated primarily to assessing the effectiveness of sterilization with dry hot air, it can also be used in the case of sterilization with UVC radiation [20,21]. *Bacillus atrophaeus* sporulation occurs as a result of the depletion of essential nutrients. This process results in extremely stress-resistant spores that can be successfully used to assess the effectiveness of sterilization with dry air, ethylene oxide, or microwaves [22]. *B. atrophaeus* spores were also used to test the effectiveness of the sterilization of air circulation systems with the use of HEPA and UVC filters [20]. Due to non-pathogenicity, ease of breeding, and at the same time, high resistance, *B. atrophaeus* spores play a fundamental role in monitoring a series of new sterilization and disinfection products [23].

The main purpose of this work was to develop a methodology for the design of portable sterilizers that can be used to sterilize various objects and surfaces, in particular for sterilizing masks with HEPA filters. As part of the work, a method of determining the spatial distribution of the radiation intensity incident on the surfaces of sterilized objects was demonstrated. The aim of the study was to determine the minimum exposure time that allows for effective deactivation of the pathogen and the maximum time, which if exceeded, may cause deterioration of the filtering properties of the HEPA filters. The effectiveness of the device was checked by performing appropriate biological tests. As part of the work, microscopic tests were also carried out to assess the effects of ultraviolet radiation on the surface structure of a filter made of a polymer material.

## 2. Method

### 2.1. Construction of the Sterilizer

Commercially available emitters can be used as the UVC radiation source, for example HNS L 55 W 2G11 manufactured by OSRAM or TUV PL-L 55 W/4P HF 1CT/25 manufactured by Phillips (Figure 2). The main technical parameters for both emitters are almost the same and are as follows: electrical power consumption 55 W, dominant wavelength 254 nm, and radiant flux 17 W at a wavelength range from 200 to 280 nm.

The housing for the sterilizer was made of aluminum. This material was chosen as it is light, durable, and resistant to corrosion and UV radiation. Moreover, it is easy to be worked with and widely available in different shapes. The construction and dimensions of the completed sterilizer are presented in Figure 3. The emitters were mounted at a distance of 15 cm from each other and 15 cm from the surface to be sterilized. Figure 4 presents photos of the sterilizer.

In order to determine the required minimum and allowable maximum surface sterilization times in the UVGI method, first, the dose of ultraviolet radiation reaching the surface from the source of radiation should be determined. The proposed solution uses emitters, which emit light evenly from their entire active surface. Although the UVC radiation source consists of two parallel tubes placed close to each other Figure 2 in order to simplify calculations, both tubes will be treated as an emitter in the form of a single tube of length *L*. For this type of source, it can be assumed that the radiation density distribution is aligned along the longitudinal axis of the emitter and that on any small fragment of the emitter the radiation propagates equally in all radial directions.

### 2.2. Determining the Spatial Distribution of the Direct Radiation Intensity

Figure 5 shows a single radiator illuminating the surface to be sterilized by an element at a distance *R* from the radiator axis. The length of the active surface of the emitter is *L*.

Each unit fragment of the sterilized surface *A_i_* is illuminated from each unit cylindrical fragment of the emitter. The radiation of the emitter falls on the sterilized surface located at a distance *R_i_* from the source and at an angle of *α_i_*. Taking into account the coordinates of the unit radiation source [*x_L_*, *y_L_*, *z_L_*] and the coordinates of the unit surface [*x_i_*, 0, *z_i_*], the distance *R_i_* can be determined from the equation
(1)$$R_{i}=\sqrt{{\left(x_{i}-x_{L}\right)}^{2}+{y_{L}}^{2}+{\left(z_{i}-z_{L}\right)}^{2}}$$

The angle of incidence *α_i_* can be determined from the equation
(2)$$\alpha_{i}=\tan^{-1}\left(\frac{y_{L}}{\sqrt{{\left(x_{i}-x_{L}\right)}^{2}+{\left(z_{i}-z_{L}\right)}^{2}}}\right)$$

Assuming an even emission of radiant flux ∅ along the entire axis of the emitter of length *L*, the flux $\emptyset_{dz_{L}}^{\prime }$ emitted by a unit cylindrical fragment of the active surface of the emitter can be determined from the equation
(3)$$\emptyset_{dz_{L}}^{\prime }=\frac{\emptyset }{L}$$

Irradiance $E_{e,i,dz}^{\prime }$ received from a unit cylindrical fragment of the emitter, and reaching the unit surface *A_i_*, can be determined from
(4)$$E_{e,i,dz}^{\prime }=\frac{\emptyset_{dz_{L}}^{\prime }}{4\pi \cdot {R_{i}}^{2}}\sin\alpha_{i}$$

The total irradiance *E_e,i_* on a unit area surface is the integral of $E_{e,i,dz}^{\prime }$ over the length *L* of the active area of the emitter is
(5)$$E_{e,i}=\overset{L}{\underset{0}{\int }}E_{e,i,dz}^{\prime }\;dz_{L}$$

### 2.3. The Method of Verifying the Effectiveness of Sterilization

The studies used standard bio-indicators in the form of strips, which were inoculated with *B. atrophaeus* spores (ATCC 9372) at >10^6^ spores/strip (ACE test, Fukuzawa Shoji Co., Ltd., Yokohama, Japan). The product meets the requirements of the ISO 13485 standard regarding quality management systems in the medical devices industry. Biological indicators are supplemented with Soybean Casein Digest broth, which acts as a culture medium, enabling germination and growth of microorganisms that have survived the sterilization process. During the growth of microorganisms, enzymatic reactions take place that make the pH of the medium acidic [24]. The sterilization efficiency is assessed on the basis of the color change of the sample, from red to yellow, due to the presence of the pH indicator—phenol red.

The mask, consisting of 9 layers, was cut into pieces measuring 4 × 2 cm. The fragments consisted of 2 outer layers (layers 1 and 9) and 7 internal layers (Figure 6a). The front of the mask in some places has been provided with a stiffener, which is also a second layer. (Figure 6b). Indicator strips were placed on the surface and under the 1st, 2nd, 3rd, 4th, 5th, and 8th layer of the mask (Figure 6c). The tests were carried out on parts of the mask with and without the stiffener. The prepared samples were placed on petri dishes at a distance of 15 cm from the radiators and irradiated for 15 min. After this time, the indicator papers were placed in culture media and incubated in an incubator at 37 °C for 7 days. The procedure was performed in four replications for each layer of the mask. Additionally, control tests were carried out, where the mask fragments were not irradiated, and the UVC lamps were turned off.

## 3. Results and Discussion

Spatial distributions of the direct radiation irradiance originating from a single active emitter are shown in Figure 7. The calculations and charts were made using a program written in the LabView environment, in accordance with the theoretical assumptions and formulas presented in the chapter, Methods.

Figure 8 shows the spatial distribution of the direct radiation irradiance *E_e_* from five simultaneously operating radiation emitters.

The distance 15 cm between the emitters was chosen as a compromise between the two solutions. An arrangement of emitters very close together allows for a uniform distribution of irradiance but only on a small working area. The arrangement of emitters at some greater distance increases the size of the working area, which generally allows one to sterilize more items. Increasing the distance, however, causes the appearance of larger differences in the irradiance in poorly and heavily lit places.

The distance of 15 cm between emitters has been related to the assumed nominal distance of sterilized objects from emitters. The distance of 15 cm between the object and the emitter seems to be optimal. For currently commercially produced UVC radiation emitters, placing the object at a distance of 15 cm allows a 1 J/cm² dose to be delivered to the surface within a few minutes, which is not a long time. Increasing the distance from the source will extend this time, which seems undesirable. On the other hand, decreasing the distance will shorten the exposure time, but will increase the unevenness of the exposure. This unevenness of the exposure will be due to two reasons. First, radiation reaching the flat surface will be emitted mainly from one direction, from the nearest lamp. Radiation from other emitters will reach the surface from a very small angle. Secondly, the sterilized surface is not perfectly flat. For a fragment protruding 1 cm above the remaining surface of the object, for a distance of 5 cm from the emitter, it will mean a 20% change in distance. At a distance of 15 cm, it will be only 7%. Therefore, a distance of 15 cm between the object and emitters was adopted. In order to bring to each fragment of the sterilized surface lighting with a similar intensity but coming from different directions, the distance between the emitters was also assumed to be 15 cm.

### 3.1. Determining the Minimum Required Exposure Time

To determine the minimum time, it was assumed that the sterilized surface will be illuminated only by direct radiation. This assumes the absence of any reflection of the radiation inside the sterilizer. This corresponds to an extreme worst-case scenario. A limited possibility of reflection could, for example, result from extreme pollution of the internal surface of the device housing.

The presented spatial distributions of direct radiation (Figure 8) show that in the corners of the area where sterilization is carried out, the irradiance is the smallest and its value is equal to *E_e,MIN_* = 1.94 mW/cm². This means that to ensure a minimum dose of UVC radiation of 1 J/cm², regardless of where the sterilized item was placed, the minimum exposure time should not be less than
(6)$$T_{MIN}=1\;\frac{J}{{\mathrm{cm}}^{2}}\;/\;1.94\;\frac{\mathrm{mW}}{{\mathrm{cm}}^{2}}≅516\;s$$

### 3.2. Determining the Maximum Allowed Exposure Time

In order to determine the maximum exposure time, it was assumed that all radiation generated by all emitters will reach the surface on which the objects to be sterilized will be located without any loss. Such a case is completely unreal, but nevertheless it allows one to determine the theoretical value of the absolute maximum radiation dose. Illumination of the surface to be sterilized with all the radiation generated by the emitters would be possible under the condition of perfect lossless reflection of waves by the internal surfaces of the device housing, as well as under the condition of full absorption of radiation on the surface. For five emitters, each generating a flux of 17 W, and with the sterilizer working area of size 50 cm × 60 cm, the maximum radiation intensity *E_e,MAX_* can be determined from
(7)Ee, MAX=5·17 W50 cm·60 cm=28.33 mWcm2

So that the maximum value of the radiation dose does not cause a significant deterioration in the properties of the polymer filter material, it should be less than 100 J/cm². The exposure time should therefore not be longer than
(8)$$T_{MAX}=100\;\frac{J}{{\mathrm{cm}}^{2}}\;/\;28.33\;\frac{\mathrm{mW}}{{\mathrm{cm}}^{2}}≅3530\;s$$

### 3.3. Validation of the Calculation of the Spatial Distribution

In order to verify the correctness of the determined spatial distribution of the irradiation, a series of measurements were made in which the irradiation *E_e,measured_* was measured at several selected points of the sterilizer’s working area. Due to the symmetry of the device structure and the symmetries of the determined spatial distribution, it was decided to take measurements in only a quarter of the working area. The locations of the selected points are shown in Figure 9. The measurements were made with the HD2302 m with the LP 471 UVC probe manufactured by DeltaOhm. Considering all factors affecting measurement accuracy, the expanded uncertainty of measurement (at 0.95 confidence level) was around 10% of the measured value. The uncertainty of measurement of radiation intensity for the measuring system used during tests depends on the following factors: calibration accuracy, effectiveness of temperature compensation of indications, aging effect of the photosensitive element of the probe (which was important in the case of the measuring probe used), and accuracy of correction of the cosine sensitivity of the probe directional sensitivity. The results of the measurements are summarized in Table 1.

The measured radiation values for individual measuring points P1–P7 are smaller than the maximum limit value calculated for the ideal reflection case, and higher than the value calculated for direct radiation, i.e., for the complete absence of reflection. The measurement results show that the reflected radiation also reaches the surface on which sterilization is carried out. The amount of this radiation depends primarily on what material the sterilizer housing is made of. The sterilizer housing, assembled for the purposes of testing, was made of aluminum, because of its good mechanical properties, resistance to ultraviolet radiation, and good radiation reflecting capability. In this case, the measurements and theoretical values (Table 1) show that approximately 30% of the radiation intensity is increased by reflected and diffused radiation, thus lost radiation not directly illuminating the surface can be estimated at around 70%. As research shows [25], good results can be also achieved by placing aluminum foil on the inner walls of the sterilizer housing. Further development of the device structure may be directed at increasing the reflection coefficient inside the sterilizer.

### 3.4. Verification of the Effectiveness of Sterilization

During the measurements verifying the effectiveness of sterilization, its outer casing was removed from the sterilizer, so that only direct radiation reached the sterilized test element. This configuration corresponded to the most unfavorable conditions of the sterilization process in which the intensity of radiation was limited. The minimum dose of radiation delivered to the system should be 1 J/cm^2^, which according to the calculations, required irradiation for at least 9 min. Considering the fact that the maximum time that will not destroy the filtration properties of the polymeric material is 55 min, the tests were carried out for 15 min so that each mask could be sterilized three times. It is clear that the components could be effectively sterilized in a shorter time as there are actually additional contributing factors, such as reflection of the radiation inside the housing or placing the sample in a well-lit place.

First, a control series was performed during which all steps of the verification procedure were performed, except that the lamps emitting UVC radiation were not turned on. During this control test, no sterilization was performed. All microbial tests for the controls showed a positive result. In all samples, growth of *B. atrophaeus* caused the medium to change color from red to yellow. Unambiguous results were obtained for four repetitions, where the indicator strips were placed on the surface and under the 1st and 8th layers of the mask. The tests included fragments of the mask with and without stiffening (Figure 10a).

The main tests were then carried out in which the lamps emitting UVC radiation were turned on, i.e., the system was operating normally. In the case of sterilization of the parts of the mask without the stiffener, the test for the presence of *B. atrophaeus* was negative. The color of the culture medium remained red in all combinations of the location of the indicator strips, i.e., on the surface and under the 1st, 2nd, 3rd, 4th, 5th, and 8th layer of the mask. Clear results were obtained for four replicates in each combination. (Figure 10b). In the case of sterilization of mask fragments containing a stiffener, UVC radiation hardly penetrates through the stiffening of the mask. Indicator tests placed on the surface and under the 1st layer showed a negative result for the presence of microorganisms (Figure 10b), but the tests placed under the 2nd stiffening and 3rd layer of the mask showed a positive result (Figure 10a).

### 3.5. Laboratory Microscopy Tests of HEPA Filters

The filter consisted of several internal layers (resembling thin paper) and two outer layers. After cutting out a filter fragment with scissors, the layers could easily be separated. Scanning electron microscopy (SEM) and transmission electron microscopy (TEM) were used to assess the effect of radiation on the filter structure. SEM imaging was performed on a FEI Helios NanoLab™ 600i microscope. Fragments of the filter layers (from the outer and inner parts) with an area of 5 mm × 5 mm were glued to the microscope table with a conductive carbon tape. Samples were not further prepared (they were not sputtered with conductive material). In addition, the morphology of the inner part (single layer) was observed in a FEI Titan 80-300 TEM microscope. Before being placed in the microscope column, a fragment of the filter layer with an area of about 1 mm × 1 mm was securely sandwiched between the two halves of the folding microscopy grid.

Imaging results confirmed that the filter consists of entangled polymer fibers with free spaces between them. In the outer layer (Figure 11a), the fibers have similar thicknesses (diameter about 30 μm). The inner layers (Figure 11b) are made of polymer fibers with heterogeneous diameters ranging from 100 nm to 10 μm. After irradiation, the morphology of the fibers did not change (Figure 11c,d).

The radiation used did not damage the material structure. No changes in shape, thickness or fiber arrangement were observed. In addition, the size of the free spaces between them has not changed significantly. Particular attention was paid to the most sensitive fibers with a diameter below 1 μm. The tests confirmed that nanofibers are present in material before (Figure 12a) and after (Figure 12b) irradiation.

## 4. Conclusions

This article presents a design of a simple sterilizer allowing the deactivation of pathogenic particles by the UVGI method. Its use is an alternative solution when it is not possible to replace filtering face masks, because it allows one to restore the filters for reuse. The designed device can also be used to sterilize a fragment of the floor, bed, table, or other objects potentially covered with pathogens. Taking into account the fact that sterilization is carried out due to the presence of a very dangerous pathogen, it is required that the effectiveness of the device for the sterilization of various objects should be confirmed by separate tests. The presented device can also be used for disinfecting other types of masks. In this case, it involves the use of holders dedicated to each type of mask, allowing the mask to be properly spread and stretched so that the entire surface of the mask can be exposed.

As part of the work, theoretical simulations of surface irradiation by a set of radiators were performed. Based on them, the placement of radiators and chamber dimensions were proposed to obtain the appropriate radiation intensity on the working surface. The methodology given can be used to design a chamber of any size and any number of radiators.

As recommended by virologists, the minimum dose of UVC radiation inactivating the SARS-COV-2 pathogen should be 1 J/cm^2^. Following these assumptions, a minimum exposure time has been set, which allows one to eliminate the pathogen, and a maximum time that will not destroy the filtration properties of the polymer material in masks. These times are 9 and 55 min for the designed chamber, respectively. The recommended sterilization time is 15 min, which gives the possibility of sterilizing one mask three times.

The results of microbiological tests show that UVC radiation eliminates pathogens in all layers of the HEPA filter. Masks without stiffening or other impenetrable layers can be successfully sterilized using UVC radiation. The only limitation is the presence of a non-transparent stiffening on the face of the mask, which inhibits the penetration of radiation to the next layers. The problem may be solved by exposing the mask from both sides. Another possibility is to introduce an additional sterilization method. Possibilities are seen in the use of gaseous chlorine dioxide [26]. The authors have plans to apply for an implementation project, which would allow the implementation and beginning of production of portable sterilizers utilizing two combined methods: UVC and chemical.

Measurements of radiation intensity in the chamber showed that the value of irradiation is greater by the value resulting from reflectivity and diffuse reflectivity of rays from the upper inner surface of the chamber and the side walls. This makes it possible to conclude that using the specified minimum exposure time you will be sure of the pathogen’s deactivation.

Based on these experiences, directions for further work were determined, including research on the phenomena of reflectivity and diffuse reflectivity of radiation in the chamber and the possibility of using LED UVC emitters, closed sterilization chamber design, and a sterilization process control system.

In response to the demand reported to the authors of this article by the head of hospital infectious disease wards, a portable compact sterilizer using UVGI ultraviolet radiation was developed and manufactured. Use of this device requires some precautions. Personnel using the device must avoid prolonged exposure of exposed parts of the body, especially the face, to radiation produced in the device. This radiation can lead to burns or reduced vision [27,28].

## 5. Limitations

During the tests, the effectiveness of sterilization was checked for only one type of mask, which was provided by the state material reserve agency to all infectious hospitals in Poland. Although, in the opinion of the authors, this type of mask is most representative of the various masks used in hospitals, additional testing of other types of masks should be performed.

The sterilization efficiency was not tested for used masks; only new masks were used in the experiments.

The studies did not carry out tests with different sterilization times. The sterilizer operation was checked always for a time of approximately 15 min. This time was assumed as the recommended sterilization time.

## Figures and Tables

**Figure 1 ijerph-17-07396-f001:**
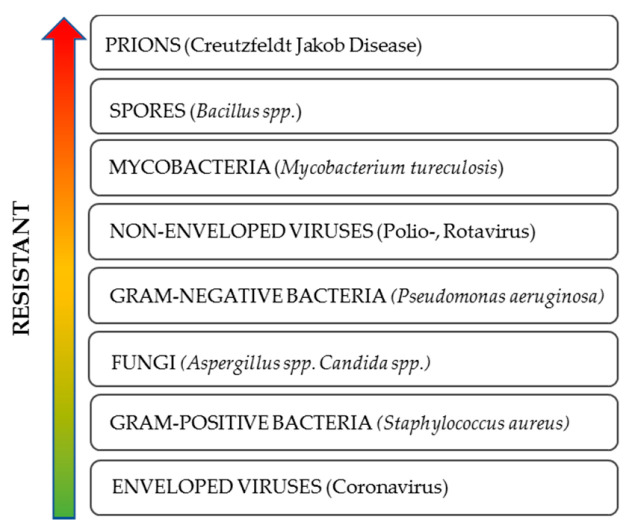
Resistance of microorganisms to sterilization and/or disinfection [16,17].

**Figure 2 ijerph-17-07396-f002:**
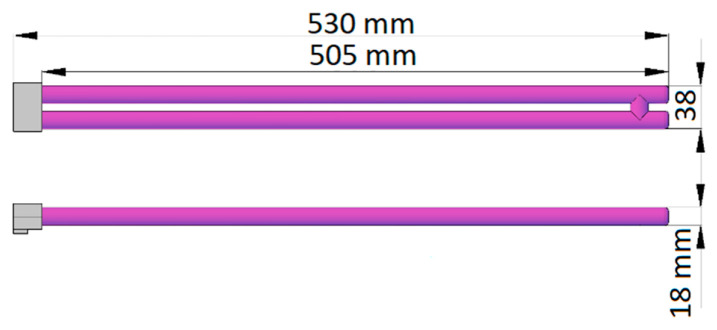
Shape and dimensions of the TUV PL-L 55 W/4P HF 1CT/25 emitter. The active surface is marked purple.

**Figure 3 ijerph-17-07396-f003:**
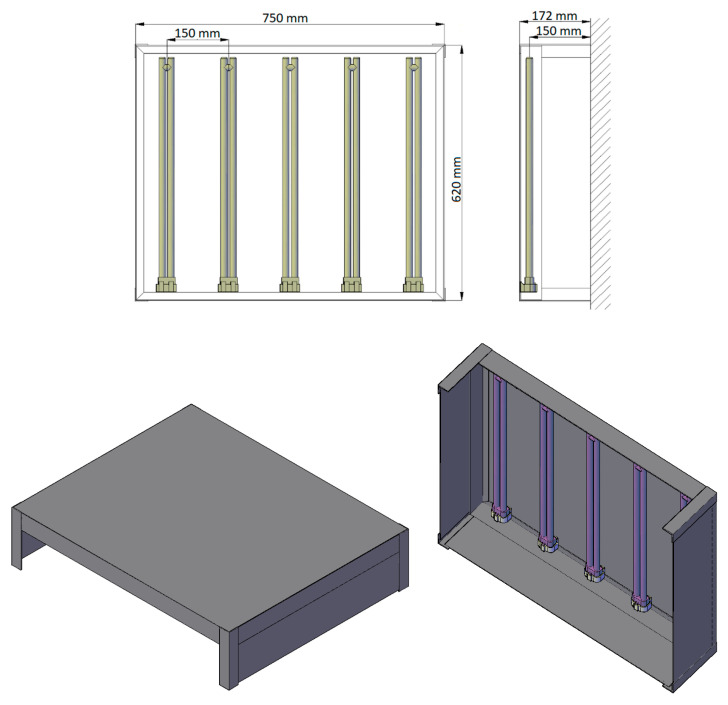
Dimensions and 3D visualization of the completed sterilizer.

**Figure 4 ijerph-17-07396-f004:**
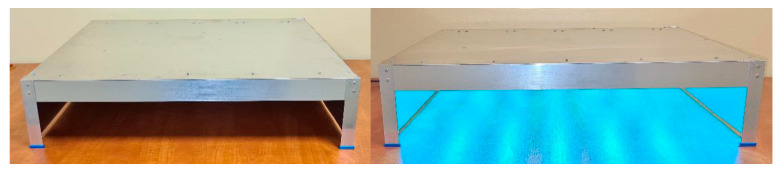
Photographs of the built sterilizer.

**Figure 5 ijerph-17-07396-f005:**
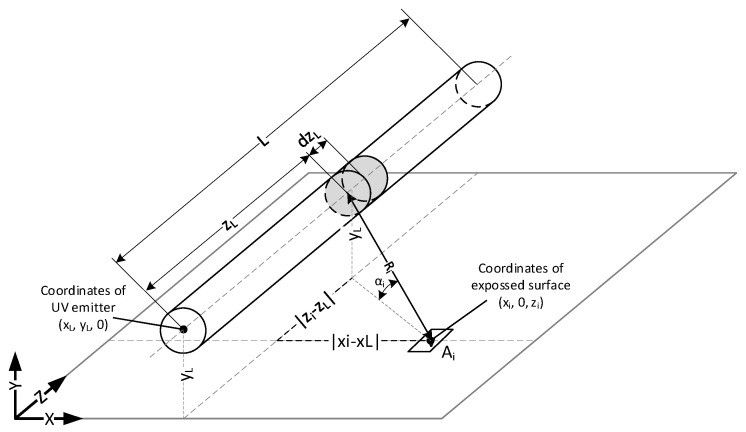
A fragment of the radiator illuminating a fragment of the sterilized surface.

**Figure 6 ijerph-17-07396-f006:**
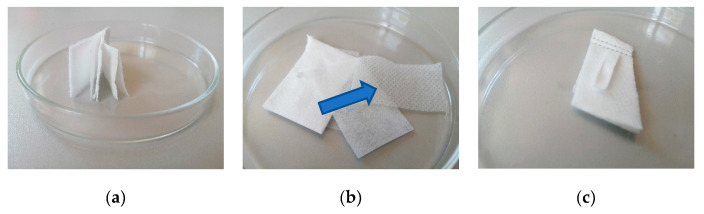
Fragment of a protective mask separated into layers (**a**); additional stiffening layer of the face mask (**b**); an indicator strip located on the surface of the mask fragment (**c**).

**Figure 7 ijerph-17-07396-f007:**
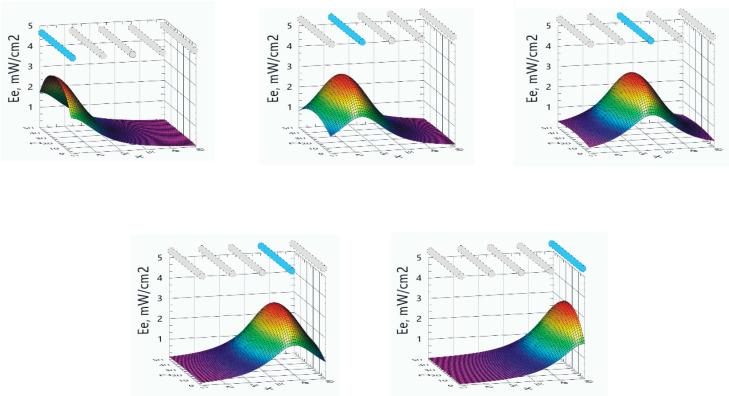
Spatial distributions of the direct radiation irradiance originating from a single active emitter.

**Figure 8 ijerph-17-07396-f008:**
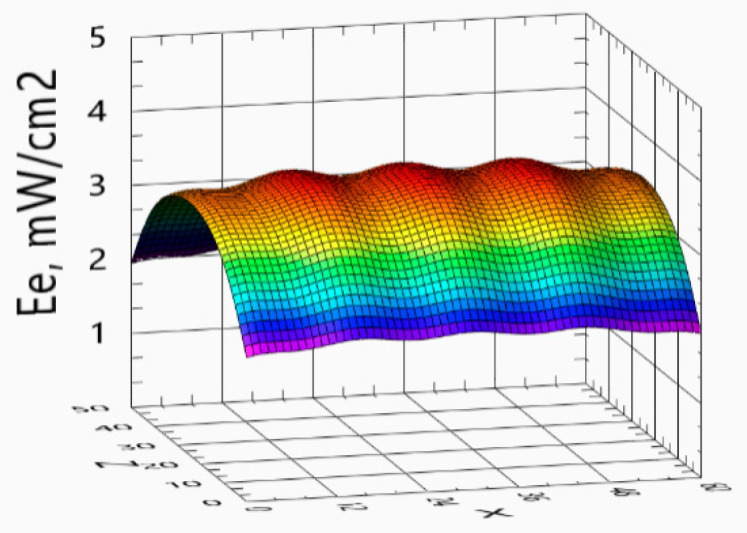
The spatial distribution of the direct radiation irradiance over the working area of the sterilizer.

**Figure 9 ijerph-17-07396-f009:**
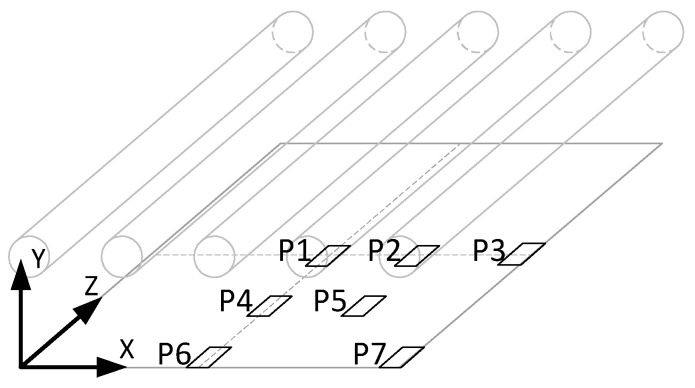
Location of irradiation measurement points during validation.

**Figure 10 ijerph-17-07396-f010:**
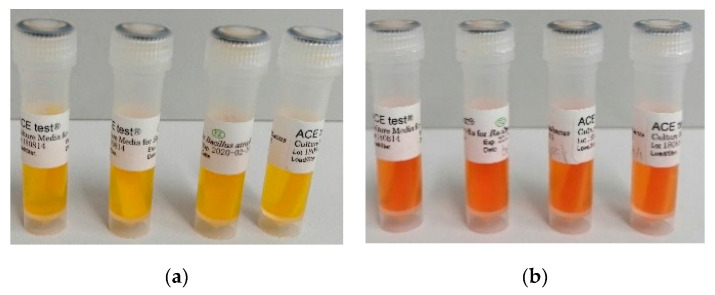
The indicator papers with *B. atrophaeus* spores (ATCC 9372) placed in culture media, after incubation. Positive result—microbial growth, (**a**). Negative result—no microbes, effective sterilization (**b**).

**Figure 11 ijerph-17-07396-f011:**
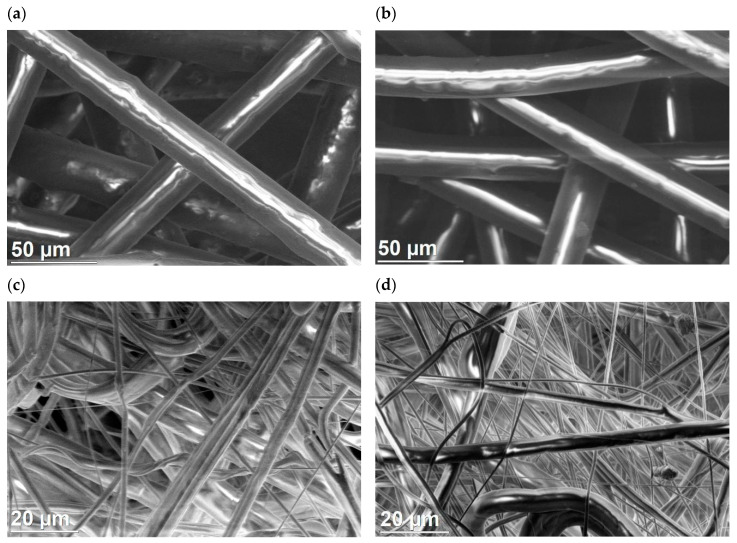
SEM images of filters. Outer layer before (**a**) and after (**b**) sterilization. Internal layer before (**c**) and after (**d**) sterilization.

**Figure 12 ijerph-17-07396-f012:**
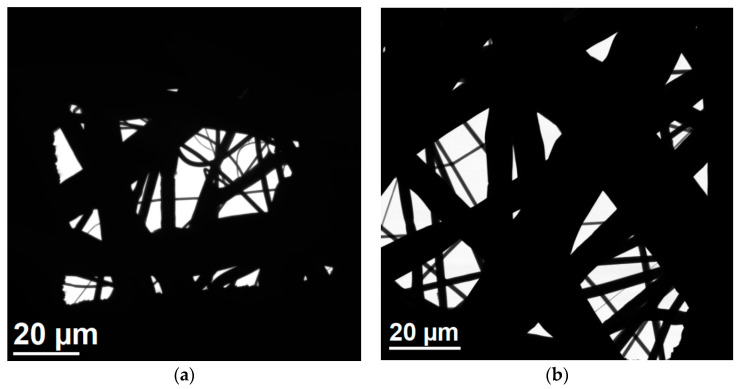
TEM images of the internal layer of the filter before (**a**) and after (**b**) sterilization.

**Table 1 ijerph-17-07396-t001:** Results of irradiation measurement during validation procedure.

Point	*E_e,calculated,ideal case_*	*E_e,measured_*	*E_e,calculated,worst case_*
	mW/cm²		
P1	28.33	4.8	3.70
P2		4.6	3.35
P3		4.6	2.13
P4		4.4	3.68
P5		4.1	3.33
P6		3.8	3.42
P7		3.3	1.94

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
