# Peer review of "The Use of UVC Irradiation to Sterilize Filtering Facepiece Masks Limiting Airborne Cross-Infection"

_ijerph, 2020, doi:10.3390/ijerph17207396_

Round 1
Reviewer 1 Report
Comments/Review: This is a useful, timely, well-written paper, where the authors have comprehensively investigated the effectiveness of UV sterilisation for FFP2/3-like masks. In fact, UV sterilisation systems have been around in the service industry for a while - such as hairdressers: https://www.ecatering.co.uk/buy/germ-x-slimline-tray-style-uvc-steriliser_795.htm?vat=vat_yes&gclid=EAIaIQobChMIj6T3zPuS7AIV8f3VCh17Cg6WEAQYASABEgJ1H_D_BwE So this is not a novel idea, but the authors give a detailed account of how they standardised the UVC radiation and examined the physical integrity of the masks afterwards - as well as checking the effective killing of the UVC using standard sterilisation inductor strips. It would have been useful to have performed the same testing/assessment of the UVC on surgical masks also - but perhaps the supplies of these masks were not as critical. The authors could also discuss other studies in this field to give more context: https://www.cebm.net/covid-19/extended-use-or-re-use-of-single-use-surgical-masks-and-filtering-facepiece-respirators-a-rapid-evidence-review/ But this does not detract from the overall quality of this article. I have no conflicts of interest with these authors or the subject/topic of this manuscript.
Author Response
Author’s respond to Reviewer comments
Reviewer:
This is a useful, timely, well-written paper, where the authors have comprehensively investigated the effectiveness of UV sterilisation for FFP2/3-like masks.
In fact, UV sterilisation systems have been around in the service industry for a while - such as hairdressers: https://www.ecatering.co.uk/buy/germ-x-slimline-tray-style-uvc-steriliser_795.htm?vat=vat_yes&gclid=EAIaIQobChMIj6T3zPuS7AIV8f3VCh17Cg6WEAQYASABEgJ1H_D_BwE . So this is not a novel idea, but the authors give a detailed account of how they standardised the UVC radiation and examined the physical integrity of the masks afterwards - as well as checking the effective killing of the UVC using standard sterilisation inductor strips.
It would have been useful to have performed the same testing/assessment of the UVC on surgical masks also - but perhaps the supplies of these masks were not as critical.
Authors answer: The authors of the article express their gratitude to the reviewer for his effort in analysing the content of the article and for his valuable comments. The authors hope that their answers will be satisfactory for the reviewer.
Reviewer:
The authors could also discuss other studies in this field to give more context: https://www.cebm.net/covid-19/extended-use-or-re-use-of-single-use-surgical-masks-and-filtering-facepiece-respirators-a-rapid-evidence-review/. But this does not detract from the overall quality of this article. I have no conflicts of interest with these authors or the subject/topic of this manuscript.
Authors answer:
The authors thank for the suggestion, the literature source indicated by the reviewer contains a lot of valuable information and guidance from the most important organizations dealing with public health problems, regarding re-use or reprocessing of single-use surgical mask and filtering facepiece respirators. Additional text has been added to introduction section, lines 43-49.

Reviewer 2 Report
The authors developed a device for sterilization of mask etc.
The device is useful especially for the limited space. It is much better if this paper can show:
1, use this device to sterilize SARS-CoV-2 directly in 15 min;
2, compare this new device with other commercial available device;
3, discuss how to commercialize this new device, including reduce the cost, etc.
Author Response
Author’s respond to Reviewer comments
Reviewer:
The authors developed a device for sterilization of mask etc. The device is useful especially for the limited space.
Authors answer:
The authors of the article express their gratitude to the reviewer for his effort in analysing the content of the article and for his valuable comments. The authors hope that their answers will be satisfactory.
Reviewer:
It is much better if this paper can show:
- Use this device to sterilize SARS-CoV-2 directly in 15 min;
Authors answer:
During the tests, the irradiation of the sterilized element lasted 15 minutes. Lines 186-187, 292-295 and 395-396.
Reviewer:
- Compare this new device with other commercial available device,
Authors answer:
The authors realize that there are commercially produced devices similar to the one presented in the article. In the case of commercially produced devices, sometimes, the effectiveness of sterilization within several dozen seconds (manufacturer's data), in addition to the multi-layer mask, raises doubts. There is no information from the manufacturer regarding research and tests confirming the effectiveness of the offered devices. There is usually only a general statement that the device uses UVC radiation for sterilization. It is difficult to find information on the amount of radiation dose in the manufacturer's information, hence it is difficult to compare the effectiveness of the action.
The aim of the article was to develop a methodology for the design, so that other scientists could use the information contained in our manuscript and successfully create more advanced constructions. The conducted research is the starting point for future research on the effectiveness of simultaneous sterilization using various methods, for example thermal, chemical and UVC radiation. The parallel use of a few selected sterilization methods is an obvious solution resulting from the limitations that characterize each of the above-mentioned methods. The thermal method may destroy the sterilized item, but it sterilizes the entire volume of the item. The use of a chemical method may cause chemical reactions between the material of the sterilized element and the chemical substance. The disadvantage of the chemical method is also the possibility of unpleasant odours. The effectiveness of the UVC method is limited by the presence of elements that block the penetration of radiation through the sterilized item.
From this perspective, it seems very interesting to develop a device that will use different methods of sterilization (either simultaneously or in a specific sequence). According to the authors, the appropriate selection of the intensity in each method will allow for quick and effective sterilization of various types of medical equipment.
For this reason, the authors did not pay much attention to comparing the presented device with commercially produced devices that look very attractive, thanks to the ergonomic and modernly designed housing.
Reviewer:
- Discuss how to commercialize this new device, including reduce the cost, etc.
Authors answer:
The authors of the article also plan to complete the research on the effectiveness of the chemical method, and then start work on the construction of a device that uses three different methods of sterilization: chemical, thermal and UVC.
The final effect of these activities is to implement and launch the production of portable and universal sterilizers. The authors have plans to apply for a project in the National Center for Research and Development), whose main goal is to support such implementation projects.
Additional text has been added to conclusions, lines 372-374.

Reviewer 3 Report
This intervention is quite timey in view of the current COVID19 pandemic. The proposed intervention, and the methodology employed, is very promising and have an impact on the accessibility of face masks. This could be of particular benefit in resource limited regions of the world where re-use of face masks could be essential, especially for healthcare workers.
This paper describes a methodology for the design of portable UV sterilisers. Also, calculations were developed to determine the minimal exposure time required for effective deactivation, and the maximum time before the integrity of the face mask is compromised. Lastly the effectiveness of the device was determined by performing biologic tests.
The methodology and results are clearly outlined.
Minor recommendations
- Line 344-345:The design could potentiallybe used to sterilize a fragment of the floor, etc. This has not been tested, and although it seems logical, this requires testing before making a firm assertion.
- Line 357-358:According to this study the recommended sterilization time is 15 minutes, we could not safely assume that it is possible to sterilize one mask three times, esp. on three different occasions, as there may be other factors over time that may affect the integrity of the mask.
- Another limitation of this study was that used masks were not tested, which may likely affect the outcome of this study. This is of concern as the presence of a non-transparent stiffening of the face mask compromised the penetration of radiation to the next layers. This may be relevant in daily clinical practice and should be reported as a limitation.
- It should be noted that the device performed well in laboratory circumstances and needs to be tested rigorously in routine clinical practice or in uncontrolled environment.
Overall, the initial results are encouraging and certainly holds significant promise should further testing duplicate these results.
Author Response
Author’s respond to Reviewer comments
Reviewer:
This intervention is quite timey in view of the current COVID19 pandemic. The proposed intervention, and the methodology employed, is very promising and have an impact on the accessibility of face masks. This could be of particular benefit in resource limited regions of the world where re-use of face masks could be essential, especially for healthcare workers.
This paper describes a methodology for the design of portable UV sterilisers. Also, calculations were developed to determine the minimal exposure time required for effective deactivation, and the maximum time before the integrity of the face mask is compromised. Lastly the effectiveness of the device was determined by performing biologic tests.
The methodology and results are clearly outlined.
Authors answer:
The authors of the article express their gratitude to the reviewer for his effort in analysing the content of the article and for his valuable comments. The authors hope that their answers will be satisfactory for the reviewer and clarify his doubts.
Reviewer:
Minor recommendations
Line 344-345:The design could potentiallybe used to sterilize a fragment of the floor, etc. This has not been tested, and although it seems logical, this requires testing before making a firm assertion.
Authors answer:
The authors fully agree with reviewer remark.
In response to the reviewer's comment, appropriate text was added, lines 352-354.
Reviewer:
Line 357-358:According to this study the recommended sterilization time is 15 minutes, we could not safely assume that it is possible to sterilize one mask three times, esp. on three different occasions, as there may be other factors over time that may affect the integrity of the mask.
Authors answer:
Indeed, by an oversight, the manuscript lacks information on the additional requirements that must be met in order for the mask to be sterilized, in order to reuse it.
Additional text was added to introduction section, lines 46-49.
Reviewer:
Another limitation of this study was that used masks were not tested, which may likely affect the outcome of this study. This is of concern as the presence of a non-transparent stiffening of the face mask compromised the penetration of radiation to the next layers. This may be relevant in daily clinical practice and should be reported as a limitation.
Authors answer:
The authors agree with the reviewer's remark. Only new masks were tested in the research.
In response to the reviewer's comment, authors added additional text to limitation section, lines 393-394.
The authors referred to the problem of the influence of non-transparent elements on sterilization efficiency in conclusions, lines 370-371. The authors pointed to the need to expose the mask in such cases from two sides.
The authors also pointed to another solution consisting in the simultaneous use of other sterilization methods, for example the gas method, line 372-373.
Information about planned actions to solve the problem has been added to the text, lines 373-375.
Reviewer:
It should be noted that the device performed well in laboratory circumstances and needs to be tested rigorously in routine clinical practice or in uncontrolled environment.
Authors answer:
The device designed by the authors has been successfully used since the beginning of the pandemic in a hospital where people infected with CoVID-19 are treated.
To date, the authors have not received any critical comments regarding the effectiveness of the device. The device has been in operation for half a year.
Reviewer:
Overall, the initial results are encouraging and certainly holds significant promise should further testing duplicate these results.
Authors answer:
Thank You.

This manuscript is a resubmission of an earlier submission. The following is a list of the peer review reports and author responses from that submission.
Round 1
Reviewer 1 Report
First of all, thank you for the effort in producing this study. This paper addressed proposed a disruptive technical solution based on the use of ultraviolet germicidial irradiation to sterilize and disinfect the surfaces of filters of breathing masks, getting them free from the new coronavirus SARS-CoV-2 along with other pathogens.
The manuscript is very concise and it is very easy to read and to understand its content. However, I still have some issues that I would like to suggest the authors to address before being published.
Introduction
Several times authors mention “CoVID-19” (the name of the disease) when they are referring to the virus “SARS-CoV-2”. This should be consistently rearranged along the text.
In lines 56-57 you mentioned “test results” from the literature, but you did not cite them. Please include the proper references.
In the final paragraph (lines 61-65) seems quite messy. In fact, the objective of this paper is not clear. Please clarify here the mais and specific objectives of this paper. Avoid using here considerations about the use of this technology/solution, like you did in the last two sentences of this paragraph.
Methods
Here you only described the construction of your device. You should describe all the other methods used for calculations (you include them in your results section). Please revise this to improve the paper organization.
In lines 67-69 you mentioned two emitters, but you only presented the one from PHILIPS in Figure 1. Was this the one used in your device? If yes, why did you mention both? Please clarify.
Why did you place the emitters separated by 15 cm from each other? Please clarify.
In Figures 1 and 2 which are the units used for the dimensions presented? Are they in mm? Please clarify in the figure captions.
Results
A considerable part of the results are actually the methods used for calculations. When you describe what you did, those are methods (e.g. lines 87-94, equations 1-5, and so on). Please consider to reorganize and move those parts to the methods section, considering Methods’ subsections to facilitate the organization.
Authors stated that “the amount of reflected radiation depends primarily on what material the sterilizer housing is made of”, but there they did discuss the use of alternative housing materials. Please consider to discuss it.
Authors declare that “the measurement results show that the reflected radiation also reaches the surface on which sterilization is carried out.” However, the paper does not showed any validation test in practice of this technology. Did you test its efficacy in real filtering facepiece masks? If yes, please include its results in order to validate this technology. Maybe this was outside the scope of this study, that is why you need to clarify the objectives of this study as I suggested before.
Discussion
Discussion is completely missing. Please consider to discuss your results and to provide some discussion on at least some of the following important questions.
- Are there advantages with this configuration (size, 15 cm spacing between emitters, aluminium as housing material,…) in comparison with alternative configurations?
- Discuss the use of this solution in real context. Is it really effective in removing pathogens/SARS-CoV-2? Will you expect different results depending on the material to disinfect? Will you expect interferences from the conditions of that material in the efficacy of this solution (e.g. humidity)?
- When using this solution in practice, filtering facepiece masks’ materials will absorb UV radiation. Will this led to effectiveness loss and to what degree?
- Is this a single use technology, i.e., how many times can you use this to disinfect a filtering piece of a facemask?
Author Response
Author’s respond to Reviewer 1 comments
Reviewer #1: First of all, thank you for the effort in producing this study. This paper addressed proposed a disruptive technical solution based on the use of ultraviolet germicidial irradiation to sterilize and disinfect the surfaces of filters of breathing masks, getting them free from the new coronavirus SARS-CoV-2 along with other pathogens.
The manuscript is very concise and it is very easy to read and to understand its content. However, I still have some issues that I would like to suggest the authors to address before being published.
Authors answer: The authors of the article express their gratitude to the reviewer for his effort in analysing the content of the article and for his valuable comments. The authors hope that their answers will be satisfactory for the reviewer and clarify his doubts. Changes introduced in the text are marked in red, removed fragments of text are marked in red and crossed out.
Reviewer #1: Introduction
Several times authors mention “CoVID-19” (the name of the disease) when they are referring to the virus “SARS-CoV-2”. This should be consistently rearranged along the text.
Authors answer:
We agree with the reviewer that the terms SARS-CoV-2 and CoVID-19 were not correctly used in the text. The term "SARS-CoV-2" is used for the virus, while the term "CoVID-19" is used for the disease it causes. In response to the reviewer's comment, appropriate changes were made to the text [lines 25, 48 and 49]
Reviewer #1: In lines 56-57 you mentioned “test results” from the literature, but you did not cite them. Please include the proper references.
Authors answer: We have introduced appropriate references to literature sources [line 64].
Reviewer #1: In the final paragraph (lines 61-65) seems quite messy. In fact, the objective of this paper is not clear. Please clarify here the mais and specific objectives of this paper. Avoid using here considerations about the use of this technology/solution, like you did in the last two sentences of this paragraph.
Authors answer: The authors agree with the reviewer's comment. The indicated fragment describing how the constructed device will be used and comments on how to operate it safely have been moved from introduction [lines 68-72] to conclusions [lines 276-280].
At the end of the Introduction section, a paragraph was added in which the objectives and scope of work were specified [lines 73-79]
“The main purpose of the work was to develop a methodology for the design of portable sterilizers that can be used to sterilize various objects and surfaces, in particular for sterilizing masks with HEPA filters. As part of the work, the method of determining the spatial distribution of the radiation intensity incident on the surfaces of sterilized objects was shown. The aim of the study was to determine the minimum exposure time that allows for effective deactivation of the pathogen and the maximum time, the exceeding of which may cause deterioration of the filtering properties of HEPA filters.”
Reviewer #1: Methods
Here you only described the construction of your device. You should describe all the other methods used for calculations (you include them in your results section). Please revise this to improve the paper organization.
Authors answer: As suggested by the reviewer, we moved the description of the method that allows determining the spatial distribution of radiation to the Method section [lines 100-134]
Reviewer #1: In lines 67-69 you mentioned two emitters, but you only presented the one from PHILIPS in Figure 1. Was this the one used in your device? If yes, why did you mention both? Please clarify.
Authors answer: We used a PHILIPS emitter in the prototype device. In the fragment of text indicated by the reviewer, two models of emitters are listed only as an example. Both emitter models are practically the same. In the opinion of the authors, providing specific models of emitters can help other people in the construction of the presented sterilizer.
Reviewer #1: Why did you place the emitters separated by 15 cm from each other? Please clarify.
Authors answer: The distance between the emitters was chosen as a compromise between the two solutions. The arrangement of emitters very close together allows for uniform distribution of radiation intensity but only on a small area. Increasing the emitter spacing increases the size of the sterilized surface, which generally allows you to sterilize more items. Increasing the spacing, however, causes the appearance of larger differences in the intensity of radiation in poorly and heavily lit places.
The distance of 15cm between emitters is related to the assumed nominal distance of sterilized objects from emitters. The distance of 15 cm between the object and the emitter seems to be optimal. For currently produced UVC radiation sources, placing the object at a distance of 15 cm allows a 1J / cm² dose to be delivered to the surface within a few minutes, which is not a long time. Increasing the distance from the source will extend this time, which seems undesirable. On the other hand, reducing the distance will shorten the exposure time, but on the other hand will increase the unevenness of the exposure. This unevenness of the exposure will be due to two reasons. First, radiation reaching the flats will reach mainly from one direction, from the nearest lamp. Radiation from other emitters will reach the surface from a very small angle. Secondly, the sterilized surface is not perfectly flat. In the case of an element protruding by 1 cm, for a distance of 5 cm from the emitter it will be 20%. However, for a 15cm distance it will be only 7%. Therefore, a distance of 15cm between the object and emitters was adopted. In order to bring to each fragment of the sterilized surface lighting with a similar intensity but coming from different directions, the distance between the emitters was also assumed to be 15cm.
Reviewer #1: In Figures 1 and 2 which are the units used for the dimensions presented? Are they in mm? Please clarify in the figure captions.
Authors answer: Figures 1 and 2 have been corrected.
Reviewer #1: Results
A considerable part of the results are actually the methods used for calculations. When you describe what you did, those are methods (e.g. lines 87-94, equations 1-5, and so on). Please consider to reorganize and move those parts to the methods section, considering Methods’ subsections to facilitate the organization.
Authors answer: The paper has been reorganized as described in the answer “Reviewer #1: Methods”
Reviewer #1: Authors stated that “the amount of reflected radiation depends primarily on what material the sterilizer housing is made of”, but there they did discuss the use of alternative housing materials. Please consider to discuss it.
Authors answer: The authors did not conduct research with other types of housing materials. They used aluminum because of its good mechanical properties, resistance to ultraviolet radiation and reflected and diffused radiation properties as suggested in article [14]. A comment was added in the article regarding the increase of the actual intensity value on the active surface [lines 211-216]. This effect reduces the exposure time of objects to radiation, which leads to a decrease in energy consumption and an increase in the number of sterilization sessions in the same time interval.
Reviewer #1: Authors declare that “the measurement results show that the reflected radiation also reaches the surface on which sterilization is carried out.” However, the paper does not showed any validation test in practice of this technology. Did you test its efficacy in real filtering facepiece masks? If yes, please include its results in order to validate this technology. Maybe this was outside the scope of this study, that is why you need to clarify the objectives of this study as I suggested before.
Authors answer: Appropriate measurements and tests were made and the results are included in the current version of the article: the estimated and measured intensity values are given in Table 1 and an appropriate comment was also added [lines 211-216]. The authors took into account that polymer materials may be damaged due to UVC radiation, thus HEPA filter degradation issues are discussed in the added chapter "Laboratory tests of HEPA filters" [lines 220-243].
Reviewer #1: Discussion
Discussion is completely missing. Please consider to discuss your results and to provide some discussion on at least some of the following important questions.
Are there advantages with this configuration (size, 15 cm spacing between emitters, aluminium as housing material,…) in comparison with alternative configurations?
Authors answer: Appropriate text has been added in the Discussion section [lines 149-169 and lines 211-216].
Reviewer #1: Discuss the use of this solution in real context. Is it really effective in removing pathogens/SARS-CoV-2? Will you expect different results depending on the material to disinfect? Will you expect interferences from the conditions of that material in the efficacy of this solution (e.g. humidity)?
Authors answer: Unfortunately, the authors of the article do not have direct access to a laboratory specializing in virus testing. Perhaps in the future such cooperation will be established and appropriate tests will be carried out. At present, the authors based their knowledge only on article [1] in which the suggestion appeared that if UVC radiation destroys pathogens from the SARS-CoV and MERS-CoV families, it also destroys the SARS-CoV-2 pathogen. In the revised version of the manuscript, the authors have supplemented the list of literature sources confirming this assumption, with each day the list is growing.
During the tests, the effectiveness of sterilization was checked for only one type of mask. In the authors' opinion, this type of mask is the most representative of a number of other object used in hospitals. However, the authors fully agree with the reviewer's suggestion that the research should be carried out on many types of materials. Therefore, the authors have provided relevant information in limitations sections [lines 281-288].
The presence of water and additional organic particles mentioned here is partially or even completely removed from filters and masks. Initially, the disadvantage of the device, consisting of a significant heating of the housing, turned into an its advantage. The emitter lamps are heating additionally the interior to a temperature of about 60°C supports the process of removing water.
Reviewer #1: When using this solution in practice, filtering facepiece masks’ materials will absorb UV radiation. Will this led to effectiveness loss and to what degree?
Authors answer: The authors took into account that polymer materials may be damaged due to UVC radiation. This may result in reduced filter performance. Relevant information cannot currently be found in the literature. For this reason, it was not described in the first version of the manuscript, but after the reviewer's comments, the authors conducted the research and included the results in the current version of the manuscript (it has been added section Laboratory tests of HEPA filters) [lines 220-243].
Reviewer #1: Is this a single use technology, i.e., how many times can you use this to disinfect a filtering piece of a facemask?
Authors answer: The degree of impact of UVC radiation on the structure properties of the polymer material was checked for only one length of exposure, it was 15 minutes. For the developed sterilizer, this length of exposure seems to be optimal, it is 50% longer than the minimum recommended exposure time. Only objects that are not dirty with blood or particles of living tissue can be sterilized, therefore it can be assumed that the masks will be sterilized only two or at most three times. Three times sterilization, whose total time will be 3x 15minutes, will not exceed the maximum sterilization time, i.e. 50 minutes. Nevertheless, the authors agree with the reviewer's suggestion that appropriate tests should be performed. Relevant information is included in the conclusion and limitations sections [lines 265-267 and lines 281-288].

Reviewer 2 Report
The paper designed a sterilization device for masks in order to realize the reuse of medical masks in such a challenging time facing SARS-CoV-2. The topic is very interesting to the public right now, however, the paper itself lacks scientific soundness and innovation. The whole paper is more like a single 'method' section of a more significant paper covering the evaluation of the device and discussion about the performance.
The reviewer would strongly suggest the authors supplement analysis regarding the evaluation of the device, other than simple calculations of the design. The evaluation could include the performance of the device dealing with various masks, at the various emitting outputs with different exposure durations. In addition, the reviewer would like to remind the authors that sterilization is not the only issue about the reuse of the masks. Increased water content and other deposited particles can adversely influence the performance of the used masks.
Author Response
Author’s respond to Reviewer 2 comments
Reviewer #2: The paper designed a sterilization device for masks in order to realize the reuse of medical masks in such a challenging time facing SARS-CoV-2. The topic is very interesting to the public right now, however, the paper itself lacks scientific soundness and innovation. The whole paper is more like a single 'method' section of a more significant paper covering the evaluation of the device and discussion about the performance.
Authors answer: The authors of the article express their gratitude to the reviewer for his effort in analysing the content of the article and for his valuable comments. The authors hope that their answers will be satisfactory for the reviewer and clarify his doubts. Changes introduced in the text are marked in red, removed fragments of text are marked in red and crossed out.
Reviewer #2: The reviewer would strongly suggest the authors supplement analysis regarding the evaluation of the device, other than simple calculations of the design. The evaluation could include the performance of the device dealing with various masks, at the various emitting outputs with different exposure durations.
Authors answer: The content of the article was supplemented with a paragraph [lines 220-243] in which the results of tests using a microscope are presented. Test results confirm that the dose of UVC radiation that is delivered in the sterilizer to the surface of disinfected objects does not cause noticeable degradation of HEPA filters.
During the tests, the effectiveness of sterilization was checked for only one type of mask, which was provided by the state material reserve agency to all infectious hospitals in Poland. In the authors' opinion, this type of mask is the most representative of a number of other masks used in hospitals. However, the authors fully agree with the reviewer's suggestion that the research should be carried out on many types of masks. Therefore, the authors have provided relevant information in limitations sections [lines 282-288].
The degree of impact of UVC radiation on the structure properties of the polymer material was checked for only one length of exposure, it was 15 minutes. For the developed sterilizer, this length of exposure seems to be optimal, it is 50% longer than the minimum recommended exposure time. Only objects that are not dirty with blood or particles of living tissue can be sterilized, therefore it can be assumed that the masks will be sterilized only two or at most three times. Three times sterilization, whose total time will be 3x 15minutes, will not exceed the maximum sterilization time, i.e. 50 minutes. Nevertheless, the authors agree with the reviewer's suggestion that appropriate tests should be performed. Relevant information is included in the limitations section [lines 282-288].
Reviewer #2: In addition, the reviewer would like to remind the authors that sterilization is not the only issue about the reuse of the masks.
Authors answer: Information about the possibility of using the sterilizer to disinfect other objects has already been placed in the abstract. In the revised version of the article, information about the possibility of disinfecting various other items has also been added to the conclusions section [lines 253-255].
Reviewer #2: Increased water content and other deposited particles can adversely influence the performance of the used masks.
The presence of water and additional organic particles mentioned here is partially or even completely removed from filters and masks. Initially, the disadvantage of the device, consisting of a significant heating of the housing, turned into an its advantage. The emitter lamps are heating additionally the interior to a temperature of about 80°C supports the process of removing water.

Reviewer 3 Report
The manuscript “ The use of ultraviolet germicidal irradiation to sterilize and disinfect the surfaces of filters of breathing masks limiting exposure to coronavirus particles in the air” for Applied Sciences proposed an advanced method to use UV irradiation to sterilize the surfaces of filters of breathing mask. However, after carefully reading the manuscript, it is not suitable for publish in the applied sciences. The comments are list below:
- The main theme of manuscript is sterilize and disinfect the surface of filters for especially coronavirus particles. However, there is no evidence or experimental result to confirm the proposed idea
- The authors control the distance of the UV light position and calculate the optimize condition of UV irradiation. In most of the case, especially for pathogens including bacteria and virus, can be de-activated after irradiation of UV light. However, the filter composition which is mainly fabricated using polymeric materials also could be damages and decrease the filter performance. Did the authors also investigate this? If not it should be also in the manuscript.
Overall, the authors claim that advanced UV irradiation for sterilize and disinfect the surface of filters but there is no strong experimental results to support the main idea. Therefore, the authors should revised the manuscript and experimental models to support the argument.
Author Response
Author’s respond to Reviewer 3 comments
Reviewer #3: The manuscript “ The use of ultraviolet germicidal irradiation to sterilize and disinfect the surfaces of filters of breathing masks limiting exposure to coronavirus particles in the air” for Applied Sciences proposed an advanced method to use UV irradiation to sterilize the surfaces of filters of breathing mask. However, after carefully reading the manuscript, it is not suitable for publish in the applied sciences.
Authors answer: The authors of the article express their gratitude to the reviewer for his effort in analysing the content of the article and for his valuable comments. The authors hope that their answers will be satisfactory for the reviewer and clarify his doubts. Changes introduced in the text are marked in red, removed fragments of text are marked in red and crossed out.
Reviewer #3: The comments are list below:
- The main theme of manuscript is sterilize and disinfect the surface of filters for especially coronavirus particles. However, there is no evidence or experimental result to confirm the proposed idea
Authors answer: In the presented article, the authors focused mainly on the technical aspects of the designed device for sterilizing masks with HEPA filters. The coronavirus pandemic surprised the whole world, and there is currently no accurate and reliable information on the sensitivity of the coronavirus to agents that harm it. At present, the authors based their knowledge only on article [1] in which the suggestion appeared that if UVC radiation destroys pathogens from the SARS-CoV and MERS-CoV families, it also destroys the SARS-CoV-2 pathogen. The current pandemic situation in our country and the lack of sufficient tests for the presence of SARS-CoV-2 virus do not allow conducting appropriate studies on the effectiveness of the method. From day to day there are new reports on the effectiveness of the deactivation of the SARS pathogen by UVC radiation. Below is a list of recent publications:
[*] Manuela Buonanno, David Welch, Igor Shuryak et al. Far-UVC light efficiently and safely inactivates airborne human coronaviruses, 27 April 2020, PREPRINT (Version 1) available at Research Square [https://doi.org/10.21203/rs.3.rs-25728/v1]
[*] Guerrini, Gian Luca. (2020). FOTOCATALISI E VIRUS. 10.13140/RG.2.2.14041.88163
[*] Stawicki, Stanislaw. (2020). Could tracheo-bronchial ultraviolet C irradiation be a valuable adjunct in the management of severe COVID-19 pulmonary infections?. 6. In press. 10.4103/IJAM.IJAM_19_20
[*] Derraik, José & Anderson, William & Connelly, Elisabeth & Anderson, Yvonne. (2020). Rapid evidence summary on SARS-CoV-2 survivorship and disinfection, and a reusable PPE protocol using a double-hit process. 10.1101/2020.04.02.20051409
The primary purpose of the research was not to test whether UVC radiation deactivates the pathogen, but to develop the device and determine its effectiveness. The required UVC dose has been determined on the base of information from published reports. In our manuscript we have provided detailed information about the construction of a simple steriliser. We performed additional microscope tests, which confirm that the dose of UVC radiation that is delivered in the sterilizer to the surface of disinfected objects does not cause noticeable degradation of HEPA filters. In response to the reviewer's suggestions, we introduced additional explanation and supplementation of the literature data to the manuscript content [lines 49-51].
Reviewer #3:
- The authors control the distance of the UV light position and calculate the optimize condition of UV irradiation. In most of the case, especially for pathogens including bacteria and virus, can be de-activated after irradiation of UV light. However, the filter composition which is mainly fabricated using polymeric materials also could be damages and decrease the filter performance. Did the authors also investigate this? If not it should be also in the manuscript.
Authors answer: The authors took into account that polymer materials may be damaged due to UVC radiation. This may result in reduced filter performance. Relevant information cannot currently be found in the literature. For this reason, it was not described in the first version of the manuscript, but after the reviewer's comments, the authors conducted the research and included the results in the current version of the manuscript (it has been added section Laboratory tests of HEPA filters) [lines 220-243].
Reviewer #3: Overall, the authors claim that advanced UV irradiation for sterilize and disinfect the surface of filters but there is no strong experimental results to support the main idea. Therefore, the authors should revised the manuscript and experimental models to support the argument.
Authors answer: The authors of the manuscript are convinced of the possibility of CoVID-19 virus inactivation using UVC radiation based on reports contained in medical publications. In the revised version of the manuscript, the authors have supplemented the list of literature sources confirming this assumption, with each day the list is growing.

Reviewer 4 Report
Reviewer's comments
This manuscript deals with an interesting and currently extremely important topic of UV inactivation of viruses and how it affects the HEPA polymer filters when subjected to disinfection. This topic is of the interdisciplinary nature demanding technical and biological scientific approach.
The article reads well but it has some major problems.
The authors have presented a sterilizer technical design with only some technical specifications regarding UV irradiation, however, in order to be considered for a research article, it should be subjected to a thorough and controlled experimental analysis and validation:
* A detailed experimental analysis of some HEPA filters when placed inside the UV chamber to experimentally assess the irradiance values across the irregular space shape of the filter
* The title mentions the Coronavirus specifically. In order to assess sterilizer effectiveness on the Coronavirus itself from the scientific point of view, the biological part is missing i.e. some real Corona viral load has to be tested in the sterilizer to analyze the virus deactivation effectiveness. Instead of the Corona virus itself some resistant laboratory biological indicators (usually spores) can be used in case their resistance is scientifically correlated with the corona virus.
* Virus deactivation on the flat surface vs. virus deactivation on HEPA filter with the irregular shape which causes various irradiation dosages over the surface.
The presented results are not sufficient as they can more be applicable to the "Materials and Methods" part of the article.
Some other important issues:
The introduction should more succinctly specify the goals and results of the research from the technical and biological point of view
Figure 7 should include specifications like coordinates of the gauge points.
Author Response
Author’s respond to Reviewer 4 comments
Reviewer #3: This manuscript deals with an interesting and currently extremely important topic of UV inactivation of viruses and how it affects the HEPA polymer filters when subjected to disinfection. This topic is of the interdisciplinary nature demanding technical and biological scientific approach.
Authors answer: The authors of the article express their gratitude to the reviewer for his effort in analysing the content of the article and for his valuable comments. The authors hope that their answers will be satisfactory for the reviewer and clarify his doubts. Changes introduced in the text are marked in red, removed fragments of text are marked in red and crossed out.
Reviewer #4: The article reads well but it has some major problems.
The authors have presented a sterilizer technical design with only some technical specifications regarding UV irradiation, however, in order to be considered for a research article, it should be subjected to a thorough and controlled experimental analysis and validation:
A detailed experimental analysis of some HEPA filters when placed inside the UV chamber to experimentally assess the irradiance values across the irregular space shape of the filter
Authors answer: During the tests, the effectiveness of sterilization was checked for only one type of mask, which was provided by the state material reserve agency to all hospitals for the treatment of infectious diseases in Poland. In the authors' opinion, this type of mask is the most representative of a number of other masks used in hospitals. However, the authors fully agree with the reviewer's suggestion that the research should be carried out on many types of masks. Therefore, the authors have provided relevant information in limitations sections [lines 282-288]. The current pandemic situation in the country does not allow this type of research, because there is even a lack of tests for people suspected of being infected. Such reliable research can be carried out in the future when the situation improves. However, it can be assumed that extending the exposure of filters to UVC radiation will allow for deeper penetration and deactivation of pathogens located there.
Reviewer #4: The title mentions the Coronavirus specifically. In order to assess sterilizer effectiveness on the Coronavirus itself from the scientific point of view, the biological part is missing i.e. some real Corona viral load has to be tested in the sterilizer to analyze the virus deactivation effectiveness. Instead of the Corona virus itself some resistant laboratory biological indicators (usually spores) can be used in case their resistance is scientifically correlated with the corona virus.
Authors answer: Unfortunately, the authors of the article do not have direct access to a laboratory specializing in virus testing. Perhaps in the future such cooperation will be established and appropriate tests will be carried out. At present, the authors based their knowledge only on article [1] in which the suggestion appeared that if UVC radiation destroys pathogens from the SARS-CoV and MERS-CoV families, it also destroys the SARS-CoV-2 pathogen. The current pandemic situation in our country and the lack of sufficient tests for the presence of SARS-CoV-2 virus do not allow conducting appropriate studies on the effectiveness of the method. From day to day there are new reports on the effectiveness of the deactivation of the SARS pathogen by UVC radiation. In response to the reviewer's suggestions, we introduced additional explanation and supplementation of the literature data to the manuscript content [lines 49-51].
Reviewer #4: Virus deactivation on the flat surface vs. virus deactivation on HEPA filter with the irregular shape which causes various irradiation dosages over the surface.
Authors answer: The use of the sterilizer for disinfecting masks involves the use of holders dedicated to each type of mask, allowing the mask to be properly spread and stretched so that the entire surface of the mask can be exposed. It can also extend exposure time to radiation without exceeding the maximum value that can damage the HEPA filter to obtain greater certainty about the deactivation of pathogen. Additional factors like temperature and a small amount of ozone can improve the situation.
Reviewer #4: The presented results are not sufficient as they can more be applicable to the "Materials and Methods" part of the article.
Authors answer: The content of the article was supplemented with a paragraph [lines 220-243] in which the results of tests using a microscope are presented. Test results confirm that the dose of UVC radiation that is delivered in the sterilizer to the surface of disinfected objects does not cause noticeable degradation of HEPA filters.
Reviewer #4: Some other important issues:
The introduction should more succinctly specify the goals and results of the research from the technical and biological point of view
Authors answer: We agree with reviewer suggestion. In the introduction a small fragment has been added indicating that all biological aspects have been taken from literature and at the moment such will not be presented in the article and discussed [lines 49-51].
Reviewer #4: Figure 7 should include specifications like coordinates of the gauge points
Authors answer: In the figure 7 has been added coordinates of the gauge points.

Reviewer 5 Report
The article presents the use of UVGI methods for deactivation of pathogens in surfaces. The reviewer recommends the following comments to be addressed by the authors:
Page 1 - Line 40: ''is confirmed by studies [1]'' - Authors are kindly requested to provide more relevant references.
Figures 5 and 6: Please state which software was used to develop the graphs.
Page 6 - Line 141: Please state if it possible to determine the practical theoretical value loss - what aspects need to be taken into account ?
Page 6 - Line 160: Please state which are these factors that affect measurement accuracy
Page 6 - Line 161: Please state how the value of 10% about the uncertainty accuracy is derived. Why not 9% or 11% ?
Page 7 - Table 1 & Lines 167-174: The value of the theoretical worst case scenario are lower than the measured ones. Please comment further on these aspects. Does this indicate that the assumptions made about the theoretical value are not valid ?
Page 7 - Conclusions: This section as it stands is very basic. Please elaborate further and expand as appropriate.
Author Response
Author’s respond to Reviewer 5 comments
Reviewer #5: The article presents the use of UVGI methods for deactivation of pathogens in surfaces. The reviewer recommends the following comments to be addressed by the authors:
Authors answer: The authors of the article express their gratitude to the reviewer for his effort in analysing the content of the article and for his valuable comments. The authors hope that their answers will be satisfactory for the reviewer and clarify his doubts. Changes introduced in the text are marked in red, removed fragments of text are marked in red and crossed out.
Reviewer #4: Page 1 - Line 40: ''is confirmed by studies [1]'' - Authors are kindly requested to provide more relevant references.
Authors answer: There are also other reports and papers on the effectiveness of the deactivation of the SARS pathogen by UVC rays. An example would be the recent publications presented below.
[*] Manuela Buonanno, David Welch, Igor Shuryak et al. Far-UVC light efficiently and safely inactivates airborne human coronaviruses, 27 April 2020, PREPRINT (Version 1) available at Research Square [https://doi.org/10.21203/rs.3.rs-25728/v1]
[*] Guerrini, Gian Luca. (2020). FOTOCATALISI E VIRUS. 10.13140/RG.2.2.14041.88163
[*] Stawicki, Stanislaw. (2020). Could tracheo-bronchial ultraviolet C irradiation be a valuable adjunct in the management of severe COVID-19 pulmonary infections?. 6. In press. 10.4103/IJAM.IJAM_19_20
[*] Derraik, José & Anderson, William & Connelly, Elisabeth & Anderson, Yvonne. (2020). Rapid evidence summary on SARS-CoV-2 survivorship and disinfection, and a reusable PPE protocol using a double-hit process. 10.1101/2020.04.02.20051409
In response to the reviewer's suggestions, we introduced additional supplementation of the literature data to the manuscript content [lines 49-51].
Reviewer #5: Figures 5 and 6: Please state which software was used to develop the graphs.
Authors answer: The calculations were made using a program written in the LabView environment, in accordance with the theoretical assumptions and formulas presented in the article.
Reviewer #5: Page 6 - Line 141: Please state if it possible to determine the practical theoretical value loss - what aspects need to be taken into account ?
Authors answer: To determine the maximum exposure time (theoretical value), it was assumed that all radiation generated by all emitters will reach the surface on which the objects to be sterilized will be located without any loss. However, it should be taken into account that the lamps radiate omnidirectional, hence only part of the radiation reaches the surface directly. This part reaches the surface without loss. The rest of the radiation reaches the surface indirectly through numerous reflections from the top and side surfaces (made of aluminium). It has been shown in [14] that a significant part of radiation in the sphere covered with aluminium foil can be reflected and diffused. In this case, however, the measurements and theoretical values (Table 1) show that approximately 30% of the radiation intensity is increased by reflected and diffused radiation. Summarize, losses radiation indirectly illuminating the surface can be estimated at around 70%.
Reviewer #5: Page 6 - Line 160: Please state which are these factors that affect measurement accuracy
Authors answer: The uncertainty of measurement of radiation intensity, for the measuring system used during tests, depends on the following factors: calibration accuracy, effectiveness of temperature compensation of indications, aging effect of the photosensitive element of the probe (which was important in the case of the measuring probe used) and accuracy of correction of the cosine sensitivity of the probe directional sensitivity. In response to the reviewer's suggestion, we introduced additional information to the manuscript content [lines 198-202].
Reviewer #5: Page 6 - Line 161: Please state how the value of 10% about the uncertainty accuracy is derived. Why not 9% or 11%?
Authors answer: The authors completely agree with the reviewer's comment. The measurement uncertainty value is a combination of many factors that affect the total resultant measurement error. The value of 10% is the value of the expanded uncertainty of measurement and was determined on the basis of standard uncertainties of the error components resulting from the impact of various factors. The method of estimating the error components was given by the manufacturer in the instrument manual. The uncertainty given is an estimated value, it is the limit value at the assumed confidence level. In response to the reviewer's suggestion, we added word “around” in the line 198.
Reviewer #5: Page 7 - Table 1 & Lines 167-174: The value of the theoretical worst case scenario are lower than the measured ones. Please comment further on these aspects. Does this indicate that the assumptions made about the theoretical value are not valid ?
Authors answer: The theoretical value for the worst case was determined using the assumption of complete absence of reflections. In this theoretical case, only direct radiation reaches the surface of the sterilized object. In real, reflected radiation also reaches the surface of the sterilized object, which is obvious. The values in the table confirm that the actual (direct and indirect) radiation intensity is greater than the (only direct) radiation intensity value for the worst case. The calculated value of radiation intensity for the worst case was used to determine the minimum sterilization time [lines 170-173].
Reviewer #5: Page 7 - Conclusions: This section as it stands is very basic. Please elaborate further and expand as appropriate
Authors answer: Conclusions has been changed on [lines 251-280]:
The article presents a design of a simple sterilizer allow deactivation of pathogenic particles by UVGI method. Its use is an alternative solution when it is not possible to replace filtering face masks, because it allows one to restore the filters for reuse. The designed device can also be used to sterilize a fragment of the floor, bed, table or other objects potentially covered with pathogen. It can be also used for disinfecting other types of masks. It this case involves the use of holders dedicated to each type of mask, allowing the mask to be properly spread and stretched so that the entire surface of the mask can be exposed.
As part of the work, theoretical simulations of surface irradiation by a set of radiators were performed. Based on them, the placement of radiators and chamber dimensions were proposed to obtain the appropriate radiation intensity on the working surface. The methodology given can be used to design a chamber of any size and any number of radiators.
As recommended by virologists, the minimum dose of UVC radiation inactivating the SARS-COV-2 pathogen should be 1J/cm2. For such assumptions, a minimum exposure time has been set, which allows to eliminate the pathogen, and a maximum time that will not destroy the filtration properties of the polymer material in masks. These times are 9 and 55 minutes for the designed chamber, respectively.
Measurements of radiation intensity in the chamber showed that the value of irradiation is greater by the value resulting from reflectivity and diffuse reflectivity of rays from the upper inner surface of the chamber and the side walls. This makes it possible to conclude that using the specified minimum exposure time you will be sure of the pathogen deactivation.
Based on the experiences, directions for further work were determined, including research on the phenomena of reflectivity and diffuse reflectivity of radiation in the chamber and the possibility of using LED UVC emitters, closed sterilization chamber design and sterilization process control system.

Round 2
Reviewer 1 Report
After the first round of revisions, authors have addressed many minor and major comments raised and have supplemented additional experimental work, by visually characterising (SEM and TEM) the surface of the HEPA filter before and after sterilization in the built device.
After careful reading the revised manuscript and authors’ answers to comments, the reviewer have concluded that the manuscript clearly improved mainly in terms of organisation. However, there are still some major flaws that should not exist after a first round of revisions, which led the reviewer to conclude that this paper is not suitable for publishing in this journal.
Most of all, there are still no experimental results proving the efficacy in sterilizing particles containing the new coronavirus (SARS-CoV-2). the reviewer agree that it might be possible to be used to sterilize SARS-CoV-2 infected filters/masks but under certain conditions which were not properly defined and tested. The reviewer also agree that it is hard to test it in real contaminated masks as authors must have access to a proper lab. However, and despite not being mentioned in the objective, authors stated it on the title of the paper, so the readers are expecting results regarding coronavirus. Thus, authors should not recommend this apparatus to be used to sterilize masks contaminated with SARS-CoV-2 without clearly prove it. The risks of its utilization without proper in vivo validation are very high.
Moreover, the tests of effectiveness were performed in only one mask, which is clearly not representative. Authors missed the chance to test more filters. One filters/mask is not representative as the authors claim. There are many types of masks/breathing filters, used by healthcare professionals and patients, built with different materials and purposes. This should be explored for the paper to be of interest to an international audience.
Lastly, authors now claim that each mask/filter can be sterilized up to three times. However, they have only tested one single sterilization. Moreover, the paper would benefit from a cost-benefit analysis of using this technology instead of buying new masks/filters, as the limited number of (re)sterilizations per mask/filter seems very low to be sustainable.
Reviewer 2 Report
The authors supplemented some surface characterization of the HEPA filter before and after sterilization. However, this kind of SEM or TEM characterization was very superficial. Instead of examining the fibers 'through your eyes', additional measurement on the properties of the fiber and the filter should be presented, such as surface area, pore distribution, etc. In a word, although the authors declare no obvious influence of the UV radiation on the fibers, the conclusion remains poorly sound. Moreover, the readers and reviewers are expecting more results of the effectiveness of the device regenerating masks, rather than simply the design process.
In addition, the authors did not address the comments from the other three reviewers.
Reviewer 3 Report
After carefully reading through the author’s respond to reviewer comment, I could not find significant improvement from the previous comments.
Most of all, there is no experimental results to prove that the UV irradiation can sterilize the coronavirus particles. Yes, in most of the case, the UV could sterilize all kinds of pathogens including other bacteria and virus and it is widely known facts. The reviewer agree that the UV could sterilize the virus but it is only possible under certain conditions and it should be confirmed properly based-on scientific experimental results and evidence to support the authors claim. However, the authors cited other journal article but did not do any experiment.
Therefore, I could not agree that the manuscript is ready to publish in this journal.
Reviewer 4 Report
The authors have made some minor improvements to the paper, however, the paper still lacks a rigorous scientific analysis as the UV sterilizer technical design is still the main focus.
Many claims given by the authors remain unproven because of the material limitations they mentioned in the manuscript. As the authors stated in their reply to the reviewer: “The current pandemic situation in the country does not allow this type of research, because there is even a lack of tests for people suspected of being infected. Such reliable research can be carried out in the future when the situation improves”.
The reviewer’s opinion is that authors should provide a thorough experimental analysis regarding the effectiveness of their technical design against the SARS-CoV-2 or any other similar virus and resubmit the paper. Any scientifically non-proven results can cause even higher damage and pose risk for the health system.